# TGFBI Facilitates Myogenesis and Limits Fibrosis in Mouse Skeletal Muscle Regeneration

**DOI:** 10.3390/ijms26189042

**Published:** 2025-09-17

**Authors:** Na Rae Park, So-Yeon Jin, Soon-Young Kim, Seung-Hoon Lee, In-San Kim, Jung-Eun Kim

**Affiliations:** 1Department of Molecular Medicine, Cell and Matrix Research Institute, School of Medicine, Kyungpook National University, Daegu 41944, Republic of Korea; nrpark85@naver.com (N.R.P.); wlsth36@gmail.com (S.-Y.J.); ksygood741@naver.com (S.-Y.K.); jsat1234@naver.com (S.-H.L.); 2BK21 FOUR KNU Convergence Educational Program of Biomedical Sciences for Creative Future Talents, Department of Biomedical Science, Kyungpook National University, Daegu 41944, Republic of Korea; 3Chemical and Biological Integrative Research Center, Biomedical Research Division, Korea Institute of Science and Technology, Seoul 02792, Republic of Korea; iskim14@kist.re.kr; 4KU-KIST Graduate School of Converging Science and Technology, Korea University, Seoul 02841, Republic of Korea

**Keywords:** transforming growth factor β-induced, myoblasts, skeletal muscle, differentiation, fusion, regeneration, fibrosis, snake venom

## Abstract

Skeletal muscles are essential for movement and support but are vulnerable to injury. Muscle regeneration relies on the extracellular matrix (ECM), which regulates key cellular processes. Transforming growth factor β-induced (TGFBI), an ECM component involved in cell adhesion, migration, and tissue development, has not been investigated in skeletal muscle regeneration. Here, we examined the role of TGFBI using *Tgfbi* knockout (KO) mice and C2C12 myoblasts. In vitro, C2C12 cells were treated with recombinant TGFBI following snake venom (SV)-induced injury, and myogenic differentiation and fusion were evaluated by quantitative real-time PCR (qRT-PCR) and Western blotting. In vivo, acute muscle injury was induced by SV injection into the tibialis anterior muscles of 12-week-old wild-type and *Tgfbi* KO mice, with regeneration assessed by histology and qRT-PCR. TGFBI was absent in uninjured muscle and C2C12 cells but was upregulated after injury. Recombinant TGFBI enhanced myogenic differentiation and restored SV-induced downregulation of myogenic and fusion markers. Although phenotypically normal under physiological conditions, *Tgfbi* KO mice exhibited impaired regeneration, characterized by persistent immature myofibers, elevated inflammatory cytokines, reduced myogenic marker expression, and increased fibrosis. These findings reveal TGFBI as a key regulator of skeletal muscle repair and a potential therapeutic target for muscle-related disorders.

## 1. Introduction

Skeletal muscle, a key component of the musculoskeletal system, facilitates body movement and maintains structural integrity [1]. Its regenerative capacity is a tightly regulated process influenced by various factors, with the extracellular matrix (ECM) emerging as a central regulator [2]. The ECM, a complex protein network, provides structural support and functions as a dynamic microenvironment that regulates cell migration, proliferation, and differentiation during muscle regeneration [3,4].

Muscle injuries resulting from trauma or underlying diseases require a robust regenerative response for functional recovery [5]. Snake venom (SV), a widely used myotoxin, induces controlled skeletal muscle damage [6]. The SV-induced muscle injury model reliably helps in investigating the complex mechanisms underlying muscle repair, activating inflammation, satellite cell responses, and ECM remodeling [6]. Snake envenomation is a serious clinical problem, often leading to extensive skeletal muscle necrosis, impaired regeneration, and long-term functional deficits. These outcomes are frequently not resolved by conventional antivenom therapies, which primarily neutralize systemic toxicity but fail to reverse local tissue damage [7,8,9]. Therefore, studying the molecular pathways involved in SV-induced injury provides not only mechanistic insights but also potential translational implications for improving treatment outcomes in patients suffering from snakebite-related muscle injury.

The transforming growth factor β-induced (*Tgfbi*) gene, also known as *βig-h3*, encodes an ECM protein expressed in various tissues, including bone, cartilage, heart, liver, and skin [10,11,12]. TGFBI regulates cell adhesion, migration, and tissue development [13,14]. As a downstream effector of TGF-β, a multifunctional cytokine involved in various cellular processes, TGFBI contributes to signaling networks regulating tissue repair [15]. However, its role in skeletal muscle, particularly in muscle regeneration, remains largely unknown.

This study aimed to investigate TGFBI expression in skeletal muscle and its role in regeneration following SV-induced injury. Using *Tgfbi* knockout (KO) mice, we elucidated the specific contribution of TGFBI to muscle repair. We examined its expression across regeneration stages and assessed its effect on key cellular events, including myoblast fusion and myogenic marker expression. Through this comprehensive approach, we focused on clarifying the interplay between ECM components, particularly TGFBI, and the regenerative processes critical for skeletal muscle function and homeostasis.

## 2. Results

### 2.1. TGFBI Enhances Differentiation and Fusion of C2C12 Cells

The ECM serves as a structural scaffold and dynamic regulator of muscle regeneration and homeostasis. Changes in its composition or signaling can significantly impact muscle function, regeneration, and disease progression. To investigate the role of TGFBI in myoblasts, we first examined its endogenous expression during C2C12 cell differentiation. As expected, MYOD expression peaked at day 0 and declined thereafter, MYOGENIN expression increased significantly by day 2, and myosin heavy chain (MyHC) expression gradually increased from day 2 to day 6. These expression patterns are consistent with the established sequence of myogenic differentiation. As TGFBI is a secreted protein, we examined its expression in both cell lysates and culture medium. In the cell lysates, TGFBI protein levels were highest at day 0 and progressively decreased during differentiation. Notably, even at day 0, the signal was relatively weak and only became visible after prolonged exposure during Western blotting, indicating that basal intracellular expression of TGFBI is minimal in C2C12 cells under normal conditions and decreases further during differentiation (Figure 1A). In the culture medium, TGFBI was also clearly detectable at day 0 and remained present at days 2 and 4, though at lower levels, confirming its secretion and subsequent downregulation during differentiation (Appendix A). Given that TGFBI is a secreted ECM protein expressed in skeletal muscles (Appendix A) and may function in early stages of muscle development, we treated C2C12 cells with varying concentrations of recombinant TGFBI to assess cytotoxicity. Doses up to 50 ng/mL showed no adverse effect on C2C12 viability (Figure 1B) and were used in subsequent experiments. Recombinant TGFBI significantly increased the number of MyHC-positive myotubes and enhanced the fusion index compared to controls (Figure 1C), indicating its role in promoting C2C12 myogenic differentiation and fusion.

### 2.2. TGFBI Rescues SV-Induced Cell Damage in C2C12 Cells

Given the role of TGFBI in cell adhesion and migration and its relevance to ECM function during regeneration [16,17,18], we investigated its involvement in muscle repair under injury conditions. Injury was induced in C2C12 cells using 7 µg/mL SV to stimulate regeneration in vitro. SV treatment significantly increased *Tgfbi* mRNA expression (Figure 2A) and the number of TGFBI-positive cells (Figure 2B) compared to controls. It also markedly reduced the expression of myogenic markers *Myod*, *Myogenin*, and *Myh3*, which are typically upregulated during differentiation (Figure 2C). However, recombinant TGFBI significantly restored their expression (Figure 2C). Similarly, SV reduced the expression of myoblast fusion markers myomaker (*Mymk*) and myomixer (*Mymx*) (Figure 2C), which was also rescued by TGFBI treatment (Figure 2C). These results indicate that TGFBI promotes myogenic differentiation and fusion during regeneration after cellular injury.

### 2.3. Tgfbi KO Mice Exhibit No Morphological Abnormalities in Skeletal Muscle Under Physiological Conditions

Since TGFBI is upregulated following SV-induced injury and promotes regeneration in vitro, we examined its in vivo role using *Tgfbi* KO mice. To assess muscle physiology, we evaluated body composition and muscle morphology. From 4 to 12 weeks, *Tgfbi* KO mice had significantly lower body weight and lean mass than wild-type (WT) mice (Figure 3A). Fat mass was also reduced at 4 and 6 weeks but showed no significant differences after 8 weeks (Figure 3A). However, when normalized to body weight, lean and fat mass ratios were similar between genotypes (Figure 3B). Similarly, *Tgfbi* KO mice exhibited reduced hindlimb muscle mass, but the muscle-to-body weight ratio remained unchanged (Figure 3C). Histological analysis of the tibialis anterior (TA) muscles at 12 weeks revealed no morphological abnormalities based on hematoxylin and eosin (H & E) staining, fiber cross-sectional area, or the cross-sectional area distribution of muscle fibers (Figure 3D). These results suggest that TGFBI is not essential for maintaining normal muscle morphology under physiological conditions.

### 2.4. TGFBI Is Induced During Early Muscle Regeneration

To evaluate TGFBI expression during in vivo muscle regeneration, we induced acute injury by injecting SV into the TA muscles of 12-week-old mice. In WT mice, *Tgfbi* mRNA expression was sharply upregulated following muscle injury, peaking at 3 days post-injection (dpi) and returning to near-baseline at 7 dpi (Figure 4A). TGFBI protein levels increased in both interstitial areas and muscle fibers at 3 dpi, peaked at 7 dpi, and remained elevated thereafter (Figure 4B). As expected, TGFBI expression was absent in *Tgfbi* KO mice (Figure 4B). By 21 dpi, *Tgfbi* KO mice displayed an altered cross-sectional area distribution of muscle fibers, with more small fibers (400 to 2400 μm^2^) and fewer large fibers (3600 to >4000 μm^2^) than WT mice (Figure 4C). *Tgfbi* KO mice also had significantly more fibers with a single central nucleus—a marker of muscle regeneration or dysfunction [19]—particularly at 7 dpi (Figure 4D). These results indicate that TGFBI is upregulated early during muscle regeneration and supports proper myofiber maturation after injury.

### 2.5. TGFBI Suppresses Fibrosis by Modulating Inflammatory Responses During Muscle Regeneration

To understand the impaired muscle regeneration in *Tgfbi* KO mice, we analyzed the expression of inflammatory mediators and myogenic markers at 3 dpi when TGFBI expression peaks in WT mice. Although TGFBI expression was absent in *Tgfbi* KO mice, the expression of *transforming growth factor beta 1* (*Tgfb1*), a known upstream regulator, was comparable between genotypes (Figure 5A). However, pro-inflammatory cytokines *interleukin 1 beta* (*Il1b*) and *interleukin 6* (*Il6*) and the pan-macrophage marker *Cd68* were significantly elevated in *Tgfbi* KO mice compared to WT mice (Figure 5B,C), while the expression of the M2 macrophage marker *Cd163* remained unchanged (Figure 5C). At 7 dpi, *Tgfbi* KO mice showed reduced expression of the satellite cell marker *Pax7* and the early myogenesis regulator *Myf5*, with *Myod* and *Myogenin* significantly downregulated (Figure 5D). The fusion markers *Mymk* and *Mymx* were also markedly downregulated in *Tgfbi* KO mice (Figure 5E), suggesting the accumulation of immature, non-fused myofibers. Quantitative analysis of Picro Sirius Red-stained fibrotic tissue revealed a significantly larger fibrotic ECM area (red-stained interstitial ECM) in regenerating TA muscle of *Tgfbi* KO mice compared to WT mice (Figure 5F). These results indicate that TGFBI deficiency promotes inflammation, impairs myogenesis, and increases fibrosis, highlighting its essential role in orchestrating balanced muscle regeneration.

## 3. Discussion

In this study, we investigated the role of TGFBI in skeletal muscle regeneration after acute injury. Although not endogenously expressed during normal C2C12 myogenic differentiation or in uninjured skeletal muscle, recombinant TGFBI significantly enhanced myoblast fusion and differentiation. Notably, TGFBI expression increased during the early phases of muscle regeneration. In *Tgfbi* KO mice, its absence delayed early regeneration, as indicated by a higher number of muscle fibers with a single central nucleus. Moreover, *Tgfbi* KO mice exhibited heightened inflammation, with elevated *Il1b*, *Il6*, and *Cd68* expression. Since controlled inflammation is crucial for muscle regeneration, this excessive inflammation likely impaired tissue repair and promoted fibrosis at later stages. Using in vitro C2C12 myoblast models and in vivo studies with *Tgfbi* KO mice, we demonstrated that TGFBI supports myogenic differentiation and fusion, regulates inflammation, and limits fibrosis during muscle regeneration (Figure 6).

The absence of morphological abnormalities in the skeletal muscle of uninjured *Tgfbi* KO mice suggests that TGFBI is not essential for normal muscle development or maintenance. This is notable given the reduced body weight and lean mass in *Tgfbi* KO mice. Similar phenotypes are observed in other ECM protein KOs, such as periostin, where reduced body size occurs without morphological abnormalities [20]. These findings suggest that while TGFBI is not required for muscle formation, it may influence overall growth through mechanisms that warrant further investigation

Our data show that TGFBI is rapidly induced following muscle injury, with *Tgfbi* mRNA expression peaking at 3 dpi and protein levels at 7 dpi. This pattern aligns with that of other injury-responsive ECM proteins, such as tenascin-C and fibronectin, which are upregulated during tissue repair, including muscle regeneration [21]. Notably, TGFBI was also detected within myofibers at 7 dpi, despite being an ECM protein, possibly due to its expression in M2 macrophages that infiltrate regenerating myofibers during muscle repair [22,23]. These findings suggest that TGFBI plays a dynamic role in muscle repair, potentially acting as a molecular switch in the transition from inflammation to regeneration.

A key finding of this study is that TGFBI promotes myogenic differentiation and fusion. Although *Tgfbi* KO mice exhibited normal baseline skeletal muscle structure, they showed impaired regeneration after SV-induced injury. At 7 dpi, these mice had fewer regenerating fibers with centralized nuclei and more fibers with a single central nucleus, indicating delayed myofiber maturation. Mechanistically, recombinant TGFBI enhanced the formation of MyHC-positive myotubes and increased the fusion index in C2C12 cells. Furthermore, TGFBI restored the SV-induced reduction in myogenic marker expression. These findings align with previous reports that specific ECM proteins promote myogenic differentiation [24]. While our in vitro findings using C2C12 myoblasts provide valuable insights into the role of TGFBI in myogenic differentiation and recovery from SV-induced injury, their murine origin limits the translational relevance of the findings. Human myoblasts have recently been used to demonstrate venom-induced impairments in viability, migration, and differentiation, thereby making them a more clinically relevant model [25]. We emphasize that incorporating human myoblast systems in future studies would strengthen the clinical relevance and therapeutic applicability of our findings for SV-induced muscle injury. Additionally, the pro-myogenic effect of TGFBI likely involves integrin-mediated signaling, as it contains RGD motifs that bind to various integrin receptors [26]. Integrins are crucial for myoblast differentiation and fusion, acting as mechanotransducers that convert ECM cues into intracellular signals [27]. Future studies should identify the specific integrin receptors and downstream pathways involved in TGFBI-mediated myogenesis. In this study, we primarily focused on characterizing the novel role of TGFBI in muscle regeneration through the regulation of myogenic differentiation, inflammation, and fibrosis following SV-induced injury. Indeed, many venom-derived enzymes, such as metalloproteases, can degrade ECM proteins and disrupt the structural scaffold required for effective regeneration. Given that TGFBI is itself an ECM protein, it is plausible that it may modulate ECM remodeling or act protectively against venom-induced degradation. Therefore, exploring how TGFBI interacts with or protects ECM components would be an important direction for future research.

Another key finding is the heightened inflammatory response observed in *Tgfbi* KO mice. The elevated *Il1b* and *Il6* expression and increased presence of macrophages (indicated by *Cd68* upregulation) in *Tgfbi* KO mice suggest that TGFBI negatively regulates inflammation during muscle regeneration. This anti-inflammatory role aligns with recent evidence demonstrating that ECM proteins modulate immune cell behavior during tissue repair [28]. TGFBI may promote macrophage polarization toward an anti-inflammatory M2 phenotype, which is essential for resolving inflammation and facilitating tissue regeneration [29]. Although the M2 macrophage marker *Cd163* levels did not differ significantly, more comprehensive analyses of macrophage subtypes and functions in the presence or absence of TGFBI are needed to clarify the immunomodulatory role of TGFBI.

The increased fibrosis in regenerating muscles of *Tgfbi* KO mice supports a regulatory role for TGFBI in ECM remodeling during muscle repair. Although induced by the pro-fibrotic cytokine TGF-β [30], TGFBI may function in a negative feedback loop to limit TGF-β-mediated fibrosis, similar to other TGF-β-regulated molecules, such as Smad7 [31]. The precise mechanisms by which TGFBI regulates ECM remodeling and fibrosis warrant further investigation, particularly regarding its potential interactions with matrix metalloproteinases, their inhibitors, and other regulators of ECM turnover. Furthermore, we noted that the cross-sectional area of regenerating fibers at 21 dpi exceeded that of uninjured controls. This enlargement reflects compensatory hypertrophy during regeneration. Such hypertrophy, where regenerating muscle fibers increase in size beyond baseline levels, has been previously reported following severe muscle injury [32]. It likely serves as an adaptive response to restore muscle function after extensive tissue damage. Clinically, snakebites often result in extensive skeletal muscle necrosis, poor regeneration, and progressive fibrosis, which collectively lead to permanent disability. Our findings support a role for TGFBI in enhancing muscle regeneration by promoting myogenic differentiation and reducing inflammation and fibrosis, highlighting its therapeutic potential for improving outcomes after envenomation.

Our findings suggest several clinical implications. By promoting myogenic differentiation and limiting inflammation and fibrosis, TGFBI emerges as a promising therapeutic target for muscle disorders characterized by impaired regeneration, chronic inflammation, or fibrosis. In Duchenne muscular dystrophy, for example, persistent inflammation and progressive fibrosis severely impair muscle function [33]. Enhancing TGFBI expression or activity could help counteract these pathological features. Similarly, TGFBI-based interventions may benefit age-related sarcopenia, characterized by reduced regeneration and increased fibrosis [34]. Furthermore, TGFBI may improve functional recovery from acute muscle injury by accelerating regeneration and minimizing fibrosis. However, this study has limitations. We focused primarily on an acute injury model, and the role of TGFBI in chronic muscle disease remains unclear. Moreover, the use of global *Tgfbi* KO mice limits our ability to resolve cell type-specific effects. Conditional KO models targeting specific cell populations, such as satellite cells, myocytes, or macrophages, could provide more precise insights into the cell-specific role of TGFBI during muscle regeneration.

In conclusion, this study identifies TGFBI as a key ECM-associated protein that supports skeletal muscle regeneration after injury by promoting myogenic differentiation and fusion, regulating inflammatory responses, and limiting fibrosis. These findings advance our understanding of ECM-mediated muscle regeneration and highlight TGFBI as a promising therapeutic target for muscle disorders characterized by impaired regeneration, elevated inflammation, or excessive fibrosis. Future studies addressing the molecular mechanisms of TGFBI and its role in human muscle disease will aid therapeutic development.

## 4. Materials and Methods

### 4.1. Animals

All animal experiments were approved by Kyungpook National University (Approval No. KNU-2022-0484, Approval date: 5 December 2022). *Tgfbi* KO mice were generated by crossing male and female *Tgfbi*^+/−^ mice [9]. Mouse genotypes were determined by PCR using tail-derived genomic DNA. The primers and PCR genotyping reactions used were described previously [9].

### 4.2. Cell Culture, MTT Assay, and SV-Induced Cell Damage

C2C12 cells were cultured for growth in Dulbecco’s modified Eagle’s medium (4.5 g/L glucose; Hyclone, Logan, UT, USA) supplemented with 10% fetal bovine serum (Hyclone, Logan, UT, USA), 2 mM L-glutamine, 100 U/mL penicillin, and 100 µg/mL streptomycin. For differentiation, cells were switched to a medium containing 2% horse serum (Life Technologies, Carlsbad, CA, USA), 2 mM L-glutamine (Gibco, Carlsbad, CA, USA), and antibiotics. For the MTT assay, C2C12 cells were seeded at 1 × 10^4^ cells/well in 48-well plates and treated with recombinant human TGFBI (R&D Systems, Minneapolis, MN, USA) at concentrations of 0, 1, 10, 50, or 100 ng/mL. After 24 h, cells were incubated with 0.5 mg/mL MTT in a culture medium for 2 h. Precipitated formazans were solubilized in dimethyl sulfoxide (Sigma-Aldrich, St. Louis, MO, USA), and absorbance was measured at 570 nm using a SPECTROstarNano (BMG Labtech, Ortenberg, Germany). For immunocytochemistry, C2C12 cells were seeded at 1.7 × 10^4^ cells/well in an 8-chamber slide (Thermo Scientific, Waltham, MA, USA) and treated with 50 ng/mL recombinant human TGFBI for 2 days of differentiation. For the analysis of *Tgfbi* expression during differentiation following SV-induced damage, C2C12 cells were seeded at 2 × 10^5^ cells/well in 6-well plates and cultured in growth medium. Prior to differentiation, cells were treated with 7 µg/mL SV for 24 h, after which SV was removed. The cells were transferred to differentiation medium and harvested every 2 days over 6 days of myogenic differentiation. To assess the protective effect of TGFBI against SV-induced cell damage, C2C12 cells were seeded at 2 × 10^5^ cells/well in 6-well plates and treated with 7 µg/mL SV for 24 h, followed by a 24 h recovery period with or without 50 ng/mL recombinant TGFBI in differentiation medium. The cells were then harvested for analysis.

### 4.3. Western Blotting

Proteins were extracted from differentiated cells using ELPIS-Biotech Protein Extraction Solution (Daejeon, Korea) containing protease inhibitors (Roche, Basel, Switzerland). Equal protein amounts were separated on Mini-PROTEAN TGX Precast Gels (Bio-Rad, Hercules, CA, USA), transferred to polyvinylidene fluoride membranes (Millipore, Burlington, MA, USA), and processed using standard Western blot protocols. Membranes were incubated with primary antibodies overnight at 4 °C, followed by appropriate secondary antibodies for 1 h at room temperature. Immunoreactive bands were visualized using a Clarity Western ECL substrate (Bio-Rad, Hercules, CA, USA). Antibody details are listed in Appendix A.

### 4.4. Immunocytochemistry

Differentiated cells were fixed with 4% paraformaldehyde (PFA, Biosesang, Yongin-si, Gyeonggi-do, Korea), permeabilized with 0.01% Triton X-100, and blocked with 1% BSA. Cells were incubated overnight at 4 °C with anti-MyHC or anti-TGFBI antibody, followed by a 30 min incubation with biotin-conjugated anti-mouse IgG antibody and then Alexa Fluor^TM^ 488 or 594 streptavidin conjugate at room temperature. Samples were mounted using ProLong Diamond Antifade Mountant containing DAPI (Invitrogen, Waltham, MA, USA). Images were acquired using a Leica DM 2500 LED optical microscope (Leica Microsystems, Wetzlar, Germany). Antibody details are listed in Appendix A.

### 4.5. Quantitative Real-Time PCR

Total RNA was extracted from TA muscles using TRIzol reagent (Life Technologies, Carlsbad, CA, USA), and cDNA was synthesized from 1 µg RNA using the RevertAid First Strand cDNA Synthesis Kit (Thermo Scientific, Waltham, MA, USA). Quantitative real-time PCR (qRT-PCR) was performed as previously described [35]. The primer sequences are listed in Appendix A.

### 4.6. Body Composition Analysis

Nuclear magnetic resonance analysis was performed biweekly on male mice aged 4 to 12 weeks to measure lean mass, fat mass, and free fluid using a minispec LF 50 (Bruker, Billerica, MA, USA). Measurements were taken without anesthesia at 10 a.m., recorded twice, and averaged.

### 4.7. Snake Venom-Induced Muscle Injury

An SV-induced muscle injury model was used to investigate muscle regeneration. Briefly, SV (Sigma-Aldrich, St. Louis, MO, USA, Cat# V5750) was dissolved in phosphate-buffered saline (PBS) at 70 μg/mL. In 12-week-old male mice, 50 µL of SV was injected into the left TA muscle, while PBS was injected into the right TA muscle as a control. Mice were anesthetized and perfused at 3, 7, 14, and 21 dpi before harvesting TA muscles for histological analysis.

### 4.8. Histological and Immunohistochemical Analyses

TA muscles were fixed in 4% PFA (Biosesang, Yongin-si, Gyeonggi-do, Korea) at 4 °C overnight, paraffin-embedded, and sectioned at 5 µm. Sections were stained with H&E and Picro Sirius Red (Abcam, Cambridge, UK) following the manufacturer’s protocols. For cross-sectional area analysis, muscle fibers were randomly selected across the entire field to minimize sampling bias, with consistent sections taken from the mid-belly region of the TA muscle. In 12-week-old uninjured mice, cross-sectional area measurements were obtained from 5 WT mice (173, 228, 208, 259, and 163 fibers per section) and 5 *Tgfbi* KO mice (130, 146, 259, 234, and 165 fibers per section). At 21 dpi, cross-sectional area was assessed from 4 WT mice (66, 48, 82, and 78 fibers per section) and 4 *Tgfbi* KO mice (102, 75, 75, and 111 fibers per section). For fibrosis analysis, Picro Sirius Red staining was performed to quantify interstitial collagen deposition between myofibers. Representative sections were taken from 5 WT and 5 *Tgfbi* KO mice (one section per mouse). The area of fibrosis, defined as red-stained interstitial ECM between myofibers, was quantified using ImageJ 1.54g software and expressed as a percentage of total tissue area. For immunohistochemistry, sections were incubated with anti-TGFBI antibody, processed using the ABC-HRP kit (Vector Laboratories, Newark, CA, USA), and visualized using the DAB substrate kit (Vector Laboratories, Newark, CA, USA) according to the manufacturer’s instructions. Tissue images were acquired using a Leica DM 100 LED microscope (Leica Microsystems, Wetzlar, Germany). Antibody details are listed in Appendix A.

### 4.9. Statistical Analysis

Data are presented as mean ± standard deviation. Differences between groups were analyzed using Student’s *t*-test, with *p* < 0.05 considered statistically significant.

## Figures and Tables

**Figure 1 ijms-26-09042-f001:**
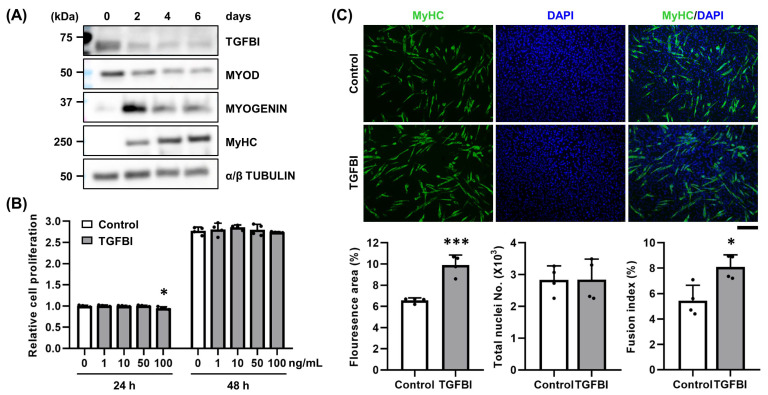
Recombinant TGFBI promotes C2C12 cell fusion. (**A**) Western blot analysis of transforming growth factor β-induced (TGFBI), MYOD, MYOGENIN, and myosin heavy chain (MyHC) expression during 6 days of C2C12 differentiation. (**B**) Cell viability was evaluated via MTT assay after treatment with recombinant TGFBI (0, 1, 10, 50, and 100 ng/mL). *n* = 4. * indicates *p* < 0.05 compared to control. h, hour. (**C**) Immunocytochemistry analysis assessed cell fusion in C2C12 cells treated with or without 50 ng/mL recombinant TGFBI for 2 days using MyHC antibody. Representative images and quantification of fluorescent area (%), total nuclei numbers (×10^3^), and fusion index (%) are shown. *n* = 4. Scale bar = 200 µm. * indicates *p* < 0.05, and *** indicates *p* < 0.001 between groups.

**Figure 2 ijms-26-09042-f002:**
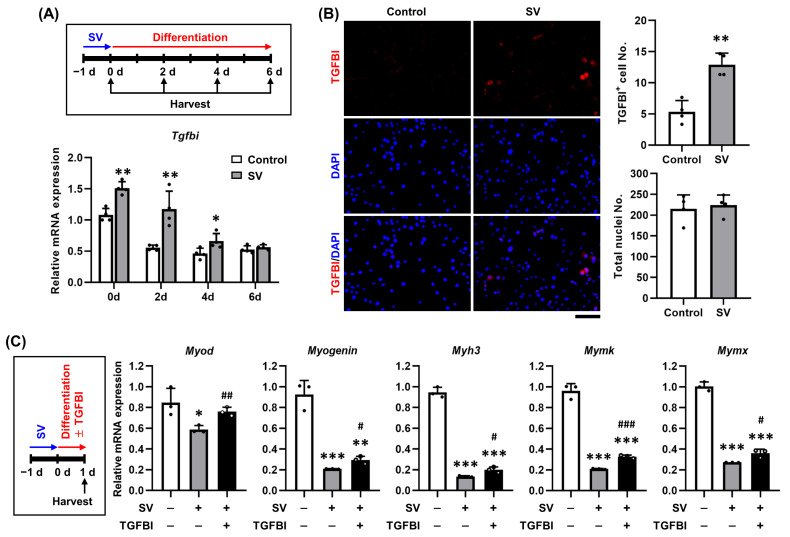
Recombinant TGFBI mitigates SV-induced damage in C2C12 cells. (**A**) Schematic of snake venom (SV) treatment in C2C12 cells and qRT-PCR analysis of *Tgfbi* mRNA expression. Prior to differentiation, cells were treated with 7 µg/mL SV for 24 h, after which SV was removed. The cells were transferred to differentiation medium and harvested every 2 days over 6 days of myogenic differentiation. qRT-PCR was performed in C2C12 cells treated with or without SV. *n* = 4. * indicates *p* < 0.05, and ** indicates *p* < 0.01 compared to control on the same day. d, day. (**B**) Immunocytochemistry of TGFBI in C2C12 cells treated with or without 7 µg/mL SV for 24 h. Representative images are shown. Quantitative data are presented as TGFBI-positive (TGFBI^+^) cell numbers and total nuclei numbers. *n* = 4. Scale bar = 100 µm. ** indicates *p* < 0.01 between groups. (**C**) Schematic of control, SV, and TGFBI treatment in C2C12 cells and qRT-PCR analysis of myogenic (*Myod*, *Myogenin*, *Myh3*) and fusion (*Mymk*, *Mymx*) markers. C2C12 cells were treated with 7 µg/mL SV for 24 h, followed by a 24 h recovery period with or without 50 ng/mL recombinant TGFBI in differentiation medium. After this, cells were harvested for analysis. *n* = 3. * indicates *p* < 0.05, and *** indicates *p* < 0.001 between control and SV. # indicates *p* < 0.05, ## indicates *p* < 0.01, and ### indicates *p* < 0.001 between SV and SV + TGFBI.

**Figure 3 ijms-26-09042-f003:**
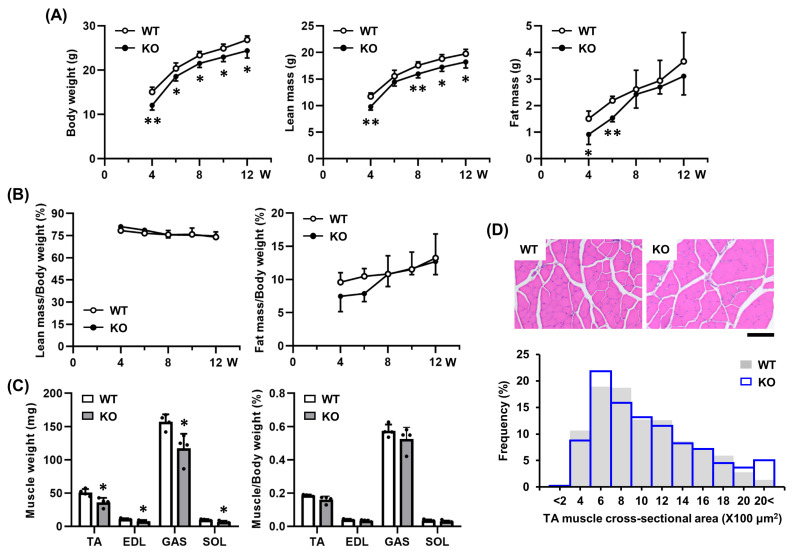
Muscle phenotype of *Tgfbi* KO mice. (**A**) Body weight, lean mass, and fat mass were measured from 4 to 12 weeks to compare wild-type (WT) and *Tgfbi* knockout (KO) mice. W, week. *n* = 5. * indicates *p* < 0.05, and ** indicates *p* < 0.01 compared to WT mice at the same age. (**B**) Lean and fat mass ratios (%) during the same period. W, week. *n* = 5. (**C**) In 12-week-old WT and *Tgfbi* KO mice, the weight of each skeletal muscle of the hindlimbs was measured and normalized to body weight (BW). *n* = 4. TA, Tibialis anterior; EDL, Extensor digitorum longus; GAS, Gastrocnemius; SOL, Soleus. * indicates *p* < 0.05 compared to WT mice at the corresponding tissue. (**D**) Fiber area morphology and size distribution were analyzed in H&E-stained TA muscles of 12-week-old WT and *Tgfbi* KO mice. Representative images are shown. Quantitative data are presented as frequency distribution (%) of TA muscle cross-sectional area. WT mice = 1031 fibers, and *Tgfbi* KO mice = 934 fibers. Scale bar = 100 µm.

**Figure 4 ijms-26-09042-f004:**
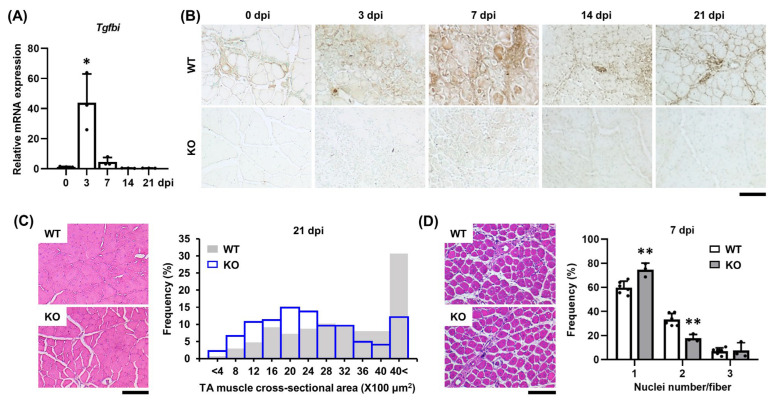
TGFBI deficiency delays SV-induced muscle regeneration. (**A**) *Tgfbi* mRNA expression in snake venom (SV)-induced tibialis anterior (TA) muscles of wild-type (WT) mice at 0, 3, 7, 14, and 21 days post-injection (dpi) using qRT-PCR. * indicates *p* < 0.05 compared to 0 dpi. (**B**) Immunohistochemistry revealed TGFBI expression in the TA muscles of WT and *Tgfbi* knockout (KO) mice following SV injection. Brown stain indicates TGFBI. Scale bar = 200 µm. (**C**) Fiber morphology and cross-sectional area distribution were analyzed in H&E-stained TA muscles at 21 dpi. Representative images are shown. Quantitative data are presented as frequency (%). WT mice = 274 fibers, *Tgfbi* KO mice = 363 fibers. Scale bar = 100 µm. (**D**) Fiber morphology and distribution by number of centralized nuclei in H&E-stained TA muscles at 7 dpi. Representative images are shown. Quantitative data are presented as the frequency distribution of central nuclei fibers (%). *n* = 5 (WT mice), *n* = 3 (*Tgfbi* KO mice). Scale bar = 100 µm. ** indicates *p* < 0.01 between genotypes.

**Figure 5 ijms-26-09042-f005:**
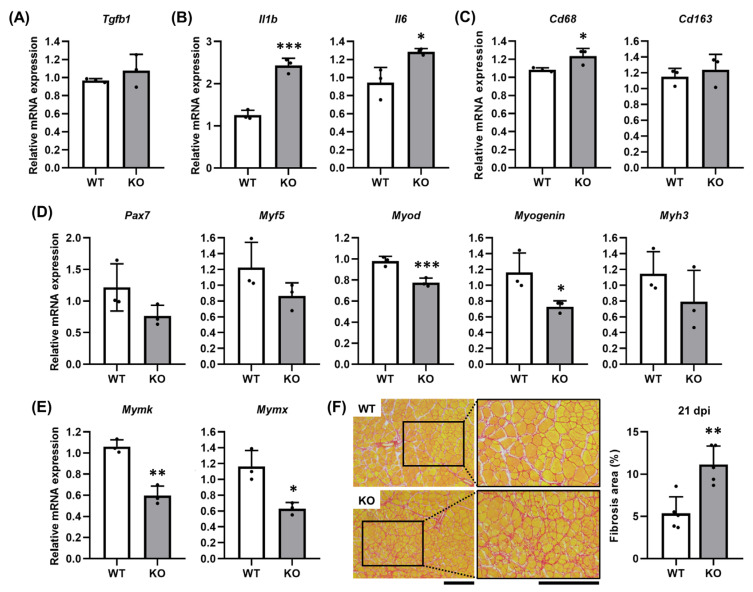
TGFBI deficiency alters marker gene expression and enhances fibrosis. (**A**–**C**) mRNA expression of *Tgfb1* (**A**), pro-inflammation cytokines (*Il1b* and *Il6*) (**B**), and macrophage markers (*Cd68* and *Cd163*) (**C**) were measured in tibialis anterior (TA) muscles of wild-type (WT) and *Tgfbi* knockout (KO) mice at 3 days post-injection (dpi) of snake venom (SV) using qRT-PCR. *n* = 3. * indicates *p* < 0.05, and *** indicates *p* < 0.001, compared to WT mice under each condition. (**D**,**E**) mRNA expression of myogenic markers (*Pax7*, *Myf5*, *Myod*, *Myogenin*, and *Myh3*) (**D**) and fusion markers (*Mymk* and *Mymx*) (**E**) were assessed at 7 dpi using qRT-PCR. *n* = 3. * indicates *p* < 0.05, ** indicates *p* < 0.01, and *** indicates *p* < 0.001, under each condition, compared to WT mice. (**F**) Fibrotic extracellular matrix (ECM) area was analyzed in Picro Sirius Red-stained TA muscles of WT and *Tgfbi* KO mice at 21 dpi. Representative images are shown. High magnification of the black boxes in the left panels is shown in the right panels. Red indicates fibrotic ECM, while yellow indicates muscle fiber and cytoplasm. Quantitative data are presented as the fibrosis area (%). *n* = 5. Scale bar = 200 µm. ** indicates *p* < 0.01 compared to WT mice.

**Figure 6 ijms-26-09042-f006:**
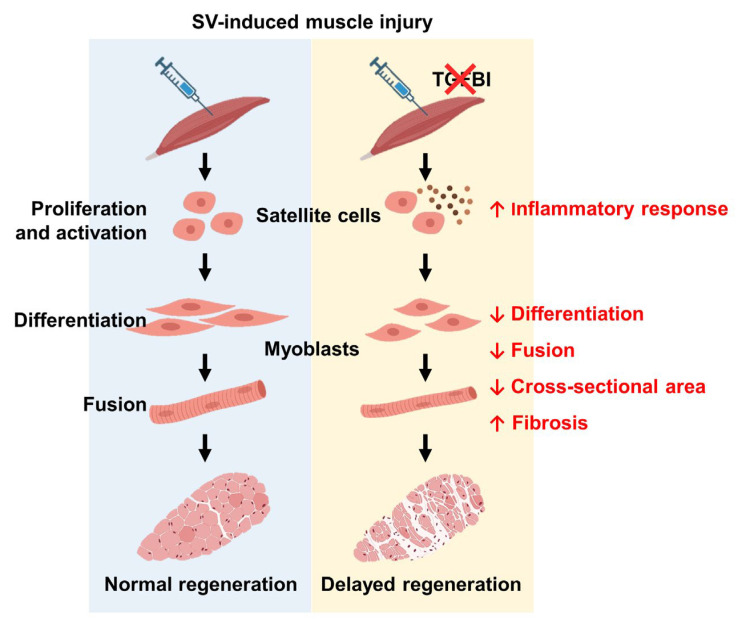
Schematic representation of the role of TGFBI in SV-induced muscle regeneration. During muscle regeneration, TGFBI promotes myoblast differentiation and fusion while modulating the inflammatory response, thereby enabling normal repair. However, TGFBI deficiency (indicated with a red cross) leads to increased expression of inflammatory genes, delayed regeneration, and enhanced fibrosis in skeletal muscle. This figure was created with BioRender (available at https://www.biorender.com/, accessed on 20 February 2025).

## Data Availability

All data generated or analyzed during this study are included in this published article [and its Appendix A Files].

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
