# Peer review of "TGFBI Facilitates Myogenesis and Limits Fibrosis in Mouse Skeletal Muscle Regeneration"

_ijms, 2025, doi:10.3390/ijms26189042_

Round 1
Reviewer 1 Report
Comments and Suggestions for Authors
The paper by Park et al. explores the role of TGFBI, an ECM component, in skeletal muscle regeneration. The authors use both in vitro (C2C12 myoblast cell line) and in vivo (Tgfbi knock-out mouse model) systems to show that Tgfbi promotes myoblasts differentiation and that its absence negatively affects recovery of skeletal muscle after injury. I have the following comments about the study:
- In vitro (C2C12), the authors can not show expression of Tgfbi during differentiation (figure 1A). However, since Tgfbi is a secreted protein, could its expression be detected maybe in the medium, rather than in the cell lysate? It should also be noted however that Tgfbi appears to be downregulated during differentiation, at least at the transcript level (figure 2B), which makes its role in C2C12 differentiation unclear.
- In the same experimental setting, while Myogenin upregulation demonstrates activation of the differentiation program (figure 1A), it is a bit strange that the Myosin isoform investigated is already expressed in proliferating conditions and not further upregulated. Maybe another antibody could be used or a different isoform could be analyzed to show proper upregulation of this important differentiation marker?
- The authors indicate using a concentration of snake venom (SV) of 1µ However, my understanding is that snake venom is a mixture of compounds (especially since authors do not provide an exact product number making it not possible to exactly establish what reagent was used), therefore it is not clear to me how the authors may have estimated the molarity of the solution.
- It is not clear how the time course experiments presented in figure 2 were performed. Cells were seeded and treated with SV for 24 hours. The treated group then received recombinant TGFBI for an additional 24 hours and was then differentiated (Materials and Methods section lines 303-306). Were recombinant TGFBI/SV present also during the differentiation or the TGFBI/SV treatment was only for the 24 hours prior differentiation? In relation to the same experiment, the histograms presented in panel D and E were collected at which stage of the time course? Also, it is not completely clear how these qRT-PCR data were analysed and normalized, as I feel that having control, SV and SV+TGFBI represented in the same graph, rather than in two separate panels, would allow better estimation of the actual effect of TGFBI on rescue of SV-induced damage.
- The cross-sectional area values presented in figures 3D and 4C should be multiplied by 100 (indeed, in the text (lines 147-148) such values are indicated as 400-2400µm2 and 3600-4000µm2). Furthermore, comparing the graph in 4C with the one presented in figure 3D it appears that at 21 dpi (figure 4C) the cross-sectional area is overall larger than baseline (figure 3D) with the majority of fibers over 4000µm2 versus 600-800µm2 in uninjured conditions (for WT). How do the authors comment on this shift towards extremely large fibers upon SV injury?
- Given the presence in snake venom of several enzymes able to alter the ECM, and since TGFBI is a secreted ECM component, it would be interesting to investigate whether, in addition to an anti-inflammatory role, TGFBI may also have a direct protective effect on ECM components upon SV injury. For instance, immunohistochemistry with markers of different ECM components could be performed in WT and Tgfbi KO muscles, or the potential protective activity of TGFBI against SV-induced ECM degradation could be assessed in vitro with simple experiments (for example, see PMID: 37372050). This information would further expand our understanding of TGFBI mechanism of action during recovery after SV injury.
Author Response
Reviewer 1
The paper by Park et al. explores the role of TGFBI, an ECM component, in skeletal muscle regeneration. The authors use both in vitro (C2C12 myoblast cell line) and in vivo (Tgfbi knock-out mouse model) systems to show that Tgfbi promotes myoblasts differentiation and that its absence negatively affects recovery of skeletal muscle after injury. I have the following comments about the study:
In vitro (C2C12), the authors cannot show expression of Tgfbi during differentiation (figure 1A). However, since Tgfbi is a secreted protein, could its expression be detected maybe in the medium, rather than in the cell lysate? It should also be noted however that Tgfbi appears to be downregulated during differentiation, at least at the transcript level (figure 2B), which makes its role in C2C12 differentiation unclear.
>> We appreciate this insightful question. To address the reviewer’s question, we repeated the western blot analysis of TGFBI during C2C2 differentiation, examining both cell lysates and culture medium, since TGFBI is a secreted protein. Consistent with the transcript data (Figure 2B), TGFBI protein levels were highest at day 0 and decreased with differentiation. In cell lysates, TGFBI levels were highest at day 0 and progressively decreased during differentiation (Figure 1A). Even at day 0, the signal was relatively weak and only became visible after prolonged exposure during western blotting. In the culture medium, TGFBI was also clearly detectable at day 0 and remained present, though at lower levels, on day 2 and 4 (Figure S1). Collectively, these findings demonstrate that endogenous TGFBI is expressed at low but detectable levels in C2C12 cells, with secretion most prominent prior to differentiation and decreasing thereafter. This expression pattern suggests that TGFBI may function mainly in the early stages of muscle development or under pathological conditions when its expression is upregulated. To further investigate TGFBI's functional role despite low endogenous expression, we treated differentiating C2C12 cells with recombinant TGFBI protein in subsequent experiments, which revealed its capacity to enhance myoblast differentiation and fusion when present at sufficient concentrations. We described this in the Results section (Lines 77-85).
In the same experimental setting, while Myogenin upregulation demonstrates activation of the differentiation program (figure 1A), it is a bit strange that the Myosin isoform investigated is already expressed in proliferating conditions and not further upregulated. Maybe another antibody could be used or a different isoform could be analyzed to show proper upregulation of this important differentiation marker?
>> We appreciate this valuable comment. To address this concern, we further the expression patterns of several key myogenic differentiation markers. MYOD expression peaked at day 0 and declined thereafter, MYOGENIN expression increased significantly by day 2, and myosin heavy chain (MyHC) expression gradually increased from day 2 to day 6. These expression patterns are consistent with the established sequence of myogenic differentiation. We have updated the Results section (Lines 74-77) to include this detailed description and clarified the interpretation of Figure 1A.
The authors indicate using a concentration of snake venom (SV) of 1 µM. However, my understanding is that snake venom is a mixture of compounds (especially since authors do not provide an exact product number making it not possible to exactly establish what reagent was used), therefore it is not clear to me how the authors may have estimated the molarity of the solution.
>> We thank the reviewer for highlighting this important issue. As correctly pointed out, snake venom is a complex mixture of proteins and enzymes, and it is not feasible to express its concentration in molarity. To clarify, we have now included the catalog number and manufacturer of the snake venom used in the Materials and Methods section (Lines 389-390). Additionally, we have revised the concentration of SV solution in the manuscript to be expressed as µg/mL (weight/volume) rather than µM, to avoid confusion and improve clarity.
It is not clear how the time course experiments presented in figure 2 were performed. Cells were seeded and treated with SV for 24 hours. The treated group then received recombinant TGFBI for an additional 24 hours and was then differentiated (Materials and Methods section lines 303-306). Were recombinant TGFBI/SV present also during the differentiation or the TGFBI/SV treatment was only for the 24 hours prior differentiation? In relation to the same experiment, the histograms presented in panel D and E were collected at which stage of the time course? Also, it is not completely clear how these qRT-PCR data were analysed and normalized, as I feel that having control, SV and SV+TGFBI represented in the same graph, rather than in two separate panels, would allow better estimation of the actual effect of TGFBI on rescue of SV-induced damage.
>> We appreciate this detailed and constructive comment. We apologize for the lack of clarity regarding the time-course experimental design.
For the analysis of Tgfbi expression during SV-induced differentiation, C2C12 cells were seeded and cultured in growth medium. Prior to differentiation, cells were treated with SV for 24 h, after which SV was removed. The cells were transferred to differentiation medium and harvested every 2 days over 6 days of myogenic differentiation. We have revised the Materials and Methods section (Lines 347-352) to more clearly describe this procedure and have updated Figure 2A (previously Figure 2A and 2B) with schematic accordingly.
To access the protective effect of TGFBI against SV-induced damage, C2C12 cells were treated with SV for 24 h, followed by a 24-hour recovery period with or without recombinant TGFBI in differentiation medium. After this, cells were harvested for analysis. We have clarified this protocol in the Materials and Methods section (Lines 352-356) and added schematic in Figure 2C. Additionally, as suggested, we have now combined the bar graphs (previously Figure 2D and 2E) into a single panel (now Figure 2C) to present the control, SV, and SV+TGFBI groups side-by-side, allowing clearer interpretation of the TGFBI rescue effect.
The cross-sectional area values presented in figures 3D and 4C should be multiplied by 100 (indeed, in the text (lines 147-148) such values are indicated as 400-2400µm2 and 3600-4000µm2). Furthermore, comparing the graph in 4C with the one presented in figure 3D it appears that at 21 dpi (figure 4C) the cross-sectional area is overall larger than baseline (figure 3D) with the majority of fibers over 4000µm2 versus 600-800µm2 in uninjured conditions (for WT). How do the authors comment on this shift towards extremely large fibers upon SV injury?
>> We thank the reviewer for carefully pointing out the discrepancy in cross-sectional area values. We have corrected the X-axis scale in Figure 3D and 4C to reflect the appropriate units.
Regarding the lager cross-sectional area at 21 dpi, we think this reflects compensatory hypertrophy during regeneration. This phenomenon, where regenerating muscle fibers increase in size beyond baseline levels, has been previously reported following severe muscle injury. Such hypertrophy likely serves as an adaptive response to restore muscle function after extensive tissue damage. We have now discussed this in the Discussion section (Lines 295-300) with a reference below.
- Dumont, N.A.; Bentzinger, C.F.; Sincennes, M.C.; Rudnicki, M.A. Satellite cells and skeletal muscle regeneration. Physiol. 2015, 5, 1027-1059.
Given the presence in snake venom of several enzymes able to alter the ECM, and since TGFBI is a secreted ECM component, it would be interesting to investigate whether, in addition to an anti-inflammatory role, TGFBI may also have a direct protective effect on ECM components upon SV injury. For instance, immunohistochemistry with markers of different ECM components could be performed in WT and Tgfbi KO muscles, or the potential protective activity of TGFBI against SV-induced ECM degradation could be assessed in vitro with simple experiments (for example, see PMID: 37372050). This information would further expand our understanding of TGFBI mechanism of action during recovery after SV injury.
>> We sincerely appreciate this valuable suggestion. As the reviewer points out, investigating whether TGFBI provides direct protection to ECM components in the context of SV-induced damage could significantly enhance our understanding of its function during muscle regeneration. While our study primarily focused on the roles of TGFBI in myogenic differentiation, inflammation, and fibrosis, we agree that its potential to preserve ECM integrity is an important avenue for future research. Given that TGFBI is itself an ECM protein, it is plausible that it may modulate ECM remodeling or act protectively against venom-induced degradation. We discussed this in the Discussion section (Lines 271-278).
The authors wish to extend our sincere thanks to the reviewer for the helpful comments.
Reviewer 2 Report
Comments and Suggestions for Authors
This study investigates the role of Transforming Growth Factor β-Induced in skeletal muscle regeneration, a previously unexplored area. Using both in vitro (C2C12 myoblasts) and in vivo (Tgfbi knockout mice) models, the authors demonstrate that TGFBI is upregulated following snake venom-induced injury and promotes myogenic differentiation and fusion. Notably, Tgfbi KO mice show impaired muscle regeneration, marked by immature myofibers, heightened inflammation, reduced expression of myogenic markers, and increased fibrosis. These findings position TGFBI as a novel and potentially actionable regulator of muscle repair, with implications for therapeutic strategies targeting muscle-related pathologies. While the manuscript presents evidences for the TGFBI’s role in muscle regeneration, several issues must be addressed before it can be considered for publication:
- Muscle injury is a frequent clinical outcome following snakebite envenomation, with significant implications for long-term disability. These sequelae are often inadequately addressed by conventional antivenom therapies. In this context, particularly in the second paragraph of the introduction, I encourage the authors to expand their discussion on venom-induced muscle damage and its downstream consequences, such as impaired regeneration and persistent functional deficits. Here are some suggestions: doi: 3389/fimmu.2020.609961, DOI: 10.3390/toxins15090530, doi: 10.1371/journal.pone.0019834 The references used in the current version are not the most appropriated for this context.
- Line 44. This is confusing. Venoms are complex mixtures with different myotoxic agents, such as PLA2, metaloproteases and cardiotoxin. It is not clear if the authors used a snake venom-derived component or the whole venom.
- Line 84. MHC is commonly used to refer to the major histocompatibility complex. However, I suggest using 'MyHC' to specifically denote myosin heavy chain.
- Even tough C2C12 are excellent models for studying muscle proliferation and differentiation. There are some limitations. Some authors are now using human myoblast to study the impact of venoms and evaluate novel therapies. This can be discussed in one paragraph of the discussion section. Here, a suggestion. https://doi.org/10.1038/s41598-024-53366-9
- I really like the schematic shown in figure 6. It summarises nicely the findings of the manuscript and its relevance.
- The discussion needs a contextualization with snakebites and the poor muscle regeneration with fibrosis following envenoming.
- It would be great if the authors have performed any for ECM components, as many venom components have proteolytic activity, which can damage the ECM scaffold and impairs regeneration.
- The authors have not described which snake venom they have used in the investigation. Snake venoms are so variable in the terms of biochemical compositions. This has huge impact on the myotoxicity of snake venoms.
- The authors have provided details regarding the analysis of histological and immunostained sections. However, it remains unclear how these sections were selected for evaluation. Additionally, the methodology for assessing fibrosis has not been described. Could the authors clarify how many muscle fibres were quantified for size measurements, and specify the criteria used for their selection?
- As the authors have used snake venom as injury model, it would be nice to include snakebites. There is an urgent need for the treatment of snakebite-induced muscle injury.
Author Response
Reviewer 2
This study investigates the role of Transforming Growth Factor β-Induced in skeletal muscle regeneration, a previously unexplored area. Using both in vitro (C2C12 myoblasts) and in vivo (Tgfbi knockout mice) models, the authors demonstrate that TGFBI is upregulated following snake venom-induced injury and promotes myogenic differentiation and fusion. Notably, Tgfbi KO mice show impaired muscle regeneration, marked by immature myofibers, heightened inflammation, reduced expression of myogenic markers, and increased fibrosis. These findings position TGFBI as a novel and potentially actionable regulator of muscle repair, with implications for therapeutic strategies targeting muscle-related pathologies. While the manuscript presents evidences for the TGFBI’s role in muscle regeneration, several issues must be addressed before it can be considered for publication:
- Muscle injury is a frequent clinical outcome following snakebite envenomation, with significant implications for long-term disability. These sequelae are often inadequately addressed by conventional antivenom therapies. In this context, particularly in the second paragraph of the introduction, I encourage the authors to expand their discussion on venom-induced muscle damage and its downstream consequences, such as impaired regeneration and persistent functional deficits. Here are some suggestions: doi: 3389/fimmu.2020.609961, DOI: 10.3390/toxins15090530, doi: 10.1371/journal.pone.0019834 The references used in the current version are not the most appropriated for this context.
>> We appreciate this valuable comment and the recommended references. Snake envenomation is a serious clinical problem, often leading to extensive skeletal muscle necrosis, impaired regeneration, and long-term functional deficits. These outcomes are frequently not resolved by conventional antivenom therapies, which primarily neutralize systemic toxicity but fail to reverse local tissue damage. Therefore, studying the molecular pathways involved in SV-induced injury provides not only mechanistic insights but also potential translational implications for improving treatment outcomes in patients suffering from snakebite-related muscle injury. We have expanded the Introduction section (Lines 47-54) to better contextualize the clinical consequences of snakebite-induced muscle damage and incorporated the suggested references below.
- Sanchez-Castro, E.E.; Pajuelo-Reyes, C.; Tejedo, R.; Soria-Juan, B.; Tapia-Limonchi, R.; Andreu, E.; Hitos, A.B.; Martin, F.; Cahuana, G.M.; Guerra-Duarte, C.; de Assis, T.C.S.; Bedoya, F.J.; Soria, B.; Chávez-Olórtegui, C.; Tejedo, J.R. Mesenchymal stromal cell-based therapies as promising treatments for muscle regeneration after snakebite envenoming. Immunol. 2021, 11, 609961.
- Sonavane, M.; Almeida, J.R.; Rajan, E.; Williams, H.F.; Townsend, F.; Cornish, E.; Mitchell, R.D.; Patel, K.; Vaiyapuri, S. Intramuscular bleeding and formation of microthrombi during skeletal muscle damage caused by a snake venom metalloprotease and a cardiotoxin. Toxins 2023, 15, 530.
- Hernández, R.; Cabalceta, C.; Saravia-Otten, P.; Chaves, A.; Gutiérrez, J.M.; Rucavado, A. Poor regenerative outcome after skeletal muscle necrosis induced by Bothrops asper venom: alterations in microvasculature and nerves. PLoS ONE 2011, 6, e19834.
- Line 44. This is confusing. Venoms are complex mixtures with different myotoxic agents, such as PLA2, metaloproteases and cardiotoxin. It is not clear if the authors used a snake venom-derived component or the whole venom.
>> We apologize for the lack of clarity. We used whole crude snake venom in this study. To clarify, we have now provided the catalog number, manufacturer, and detailed information about the snake venoms used in the Materials and Methods section (Line 389-390).
- Line 84. MHC is commonly used to refer to the major histocompatibility complex. However, I suggest using 'MyHC' to specifically denote myosin heavy chain.
>> Thank you for this insightful comment. To avoid confusion with the major histocompatibility complex, we have revised “MHC” to “MyHC” throughout the manuscript to specifically refer to myosin heavy chain.
- Even though C2C12 are excellent models for studying muscle proliferation and differentiation. There are some limitations. Some authors are now using human myoblast to study the impact of venoms and evaluate novel therapies. This can be discussed in one paragraph of the discussion section. Here, a suggestion. https://doi.org/10.1038/s41598-024-53366-9.
>> We thank the reviewer for this thoughtful suggestion. We fully agree that, although C2C12 cells are a well-established model for investigating myogenic processes, their murine origin limits their translational relevance of the findings. As suggested, we addressed this point in the Discussion section (Lines 259-266) acknowledging this limitation based on the recommended references below.
- Bin Haidar, H.; Almeida, J.R.; Williams, J.; Guo, B.; Bigot, A.; Senthilkumaran, S.; Vaiyapuri, S.; Patel, K. Differential effects of the venoms of Russell's viper and Indian cobra on human myoblasts. Rep. 2024, 14, 3184.
We reference the suggested study and note that human myoblast models have recently been used to demonstrate venom-induced impairments in viability, migration, and differentiation, thereby making them a more clinically relevant model. We emphasize that incorporating human myoblast systems in future studies would strengthen the clinically relevance and therapeutic applicability of our findings for SV-induced muscle injury.
- I really like the schematic shown in figure 6. It summarises nicely the findings of the manuscript and its relevance.
>> We thank the reviewer for this kind and encouraging feedback. We are pleased that the schematic effectively summarizes the key findings and enhances the clarity and relevance of our study.
- The discussion needs a contextualization with snakebites and the poor muscle regeneration with fibrosis following envenoming.
>> We appreciate this important comment. We have revised the Discussion (Lines 300-305) to better contextualize our findings in relation to snakebite pathology. Specifically, we now highlight that snakebites often result in extensive skeletal muscle necrosis, poor regeneration, and progressive fibrosis, which collectively lead to permanent disability. We discuss how our findings support a role for TGFBI in enhancing muscle regeneration by promoting myogenic differentiation and reducing inflammation and fibrosis, and how this may offer therapeutic potential for improving outcomes after envenomation.
- It would be great if the authors have performed any for ECM components, as many venom components have proteolytic activity, which can damage the ECM scaffold and impairs regeneration.
>> We appreciate this valuable comment. Indeed, many venom-derived enzymes, such as metalloproteases, can degrade ECM proteins and disrupt the structural scaffold required for effective regeneration. While our current study focused on characterizing the novel role of TGFBI in muscle regeneration through regulation of myogenic differentiation, inflammation, and fibrosis, we agree that exploring how TGFBI interacts with or protects ECM components would be an important direction for future research. We have described this in the Discussion section (Lines 271-278) acknowledging this limitation and highlighting the potential for future studies to assess the interaction between TGFBI and ECM integrity during regeneration following SV-induced muscle injury.
- The authors have not described which snake venom they have used in the investigation. Snake venoms are so variable in the terms of biochemical compositions. This has huge impact on the myotoxicity of snake venoms.
>> Thank you for raising this critical point. We agree that snake venom composition varies significantly by species, which directly influences its myotoxic potential. To address this, we have now included the catalog number and manufacturer about the snake venoms used in the Materials and Methods section (Line 389-390).
- The authors have provided details regarding the analysis of histological and immunostained sections. However, it remains unclear how these sections were selected for evaluation. Additionally, the methodology for assessing fibrosis has not been described. Could the authors clarify how many muscle fibres were quantified for size measurements, and specify the criteria used for their selection?
>> We appreciate this valuable comment and apologize for the lack of methodological detail. For fibrosis analysis, we used Picro Sirius Red staining to quantify interstitial collagen deposition between myofibers. Representative sections were taken from 5 WT and 5 Tgfbi KO mice (one section per mouse). The area of fibrosis, defined as red-stained interstitial ECM between myofibers, was quantified using ImageJ software and expressed as a percentage of total tissue area. For cross-sectional area analysis, muscle fibers were randomly selected across the entire field to minimize sampling bias, with consistent sections taken from the mid-belly region of the tibialis anterior muscle. In 12-week-old uninjured mice, cross-sectional area measurements were obtained from 5 WT mice (173, 228, 208, 259, and 163 fibers per section) and 5 Tgfbi KO mice (130, 146, 259, 234, and 165 fibers per section). At 21 dpi, cross-sectional area was assessed from 4 WT mice (66, 48, 82, and 78 fibers per section) and 4 Tgfbi KO mice (102, 75, 75, and 111 fibers per section). These methodological details have now been added to the Materials and Methods section (Lines 396-407).
- As the authors have used snake venom as injury model, it would be nice to include snakebites. There is an urgent need for the treatment of snakebite-induced muscle injury.
>> We appreciate this valuable comment. We agree that extending our findings to snakebite injury models would enhance the translational significance of this work. While our study utilized SV-induced injury under controlled laboratory conditions to investigate the role of TGFBI, we have now emphasized in the Discussion section (Lines 300-305) that snakebite-induced muscle injury represents an urgent unmet clinical need. We also note that future work involving actual snakebite models would help validate the therapeutic potential of TGFBI and further elucidate its function in the context of complex tissue damage.
The authors wish to extend our sincere thanks to the reviewer for the helpful comments.
Round 2
Reviewer 1 Report
Comments and Suggestions for Authors
The work by Park et al. investigates how Tgfbi exerts a protective effect on muscle regeneration upon injury induced by snake venom. The authors have made some of the suggested changes and the manuscript is now improved and clearer compared to the first version. My last suggestion would be to carry out a careful proofreading for corrections of minor language errors. As examples:
- Line 348: I feel “differentiation following SV-induced damage” would be more appropriate than “SV-induced differentiation”
- Line 352: “assess” instead of “access”
- Line 261: “the” instead of “their”
- Line 120: qRT-PCR was performed also in C2C12 cells not treated with SV, this should be clarified in the legend
- Figure 2C: For clarity, the label of the graph should read +/- TGFBI to highlight the difference in cells treatment
In conclusion, I support the publication of the corrected manuscript.
Author Response
Reviewer 1
The work by Park et al. investigates how Tgfbi exerts a protective effect on muscle regeneration upon injury induced by snake venom. The authors have made some of the suggested changes and the manuscript is now improved and clearer compared to the first version. My last suggestion would be to carry out a careful proofreading for corrections of minor language errors. As examples:
Line 348: I feel “differentiation following SV-induced damage” would be more appropriate than “SV-induced differentiation”
Line 352: “assess” instead of “access”
Line 261: “the” instead of “their”
>> We sincerely thank the reviewer for their constructive feedback and careful review of our manuscript. Following the suggestion, we have conducted a comprehensive proofreading to correct minor language errors and ensure consistency throughout the text. The grammatical and phrasing corrections are as follows:
|
Line |
Before |
After |
|
56 |
expressed in tissues |
expressed in various tissues |
|
60 |
particularly muscle regeneration |
particularly in muscle regeneration |
|
79 |
In the cell lysate |
In the cell lysates |
|
85 |
at day 2 and 4 |
at days 2 and 4 |
|
107 |
Tgfbi mRNA |
Tgfbi mRNA |
|
124 |
Schematic of SV and TGFBI treatment |
Schematic of control, SV, and TGFBI treatment |
|
165 |
displayed altered the cross-sectional area |
displayed an altered cross-sectional area |
|
188 |
Transforming growth factor |
transforming growth factor |
|
189 |
Interleukin 1 |
interleukin 1 |
|
190 |
Interleukin 6 (IL6) |
interleukin 6 (Il6) |
|
201 |
muscle regeneration |
muscle regeneration. |
|
207 |
under each condition, compared to WT mice |
compared to WT mice under each condition |
|
261 |
limits their translational relevance |
limits the translational relevance |
|
265 |
strengthen the clinically relevance |
strengthen the clinical relevance |
|
267 |
bind various integrin |
bind to various integrin |
|
272 |
through regulation |
through the regulation |
|
292 |
similar to TGF-β-regulated molecules |
similar to other TGF-β-regulated molecules |
|
298 |
This phenomenon, where |
Such hypertrophy, where |
|
299 |
Such hypertrophy likely serves |
It likely serves |
|
348 |
SV-induced differentiation |
differentiation following SV-induced damage |
|
352 |
To access |
To assess |
Line 120: qRT-PCR was performed also in C2C12 cells not treated with SV, this should be clarified in the legend
>> Thank you for pointing this out. We have revised the Figure 2 legend to: “qRT-PCR was performed in C2C12 cells treated with or without SV.”
Figure 2C: For clarity, the label of the graph should read +/- TGFBI to highlight the difference in cells treatment
>> We appreciate this valuable comment and apologize for the lack of clarity. We have revised the label of the Figure 2C graphs to clearly indicate the difference in treatment.
In conclusion, I support the publication of the corrected manuscript.
>> We would like to express our sincere gratitude once again for the reviewer’s thoughtful evaluation and encouraging support for the publication of our manuscript. All constructive suggestions have greatly improved the clarity and overall quality of our work.
Reviewer 2 Report
Comments and Suggestions for Authors
The authors have now incorporated relevant information regarding the venom used as well as details of the approaches used in the section analyses in the methodological section. The study has been better contextualized and the discussion is now more clearly framed in relation to snake venom and snakebite. I find the results interesting, and I believe that, in this context, the work is now better justified. Importantly, it also opens the door for future exploration in the development of snakebite treatments.
Author Response
Reviewer 2
The authors have now incorporated relevant information regarding the venom used as well as details of the approaches used in the section analyses in the methodological section. The study has been better contextualized and the discussion is now more clearly framed in relation to snake venom and snakebite. I find the results interesting, and I believe that, in this context, the work is now better justified. Importantly, it also opens the door for future exploration in the development of snakebite treatments.
>> We sincerely thank the reviewer for this thoughtful and encouraging feedback. Your comments helped us to better contextualize our study and frame the discussion within the broader field of snake venom and snakebite treatment research. We are especially encouraged by your view that our work opens new opportunities for future exploration in this important field. Thank you again for your valuable insights and supportive evaluation.